# Organizational Risk Factors for Aircrew Health: A Systematic Review of Observational Studies

**DOI:** 10.3390/ijerph20043401

**Published:** 2023-02-15

**Authors:** Elaine Cristina Marqueze, Erika Alvim de Sá e Benevides, Ana Carolina Russo, Mariana Souza Gomes Fürst, Rodrigo Cauduro Roscani, Paulo Cesar Vaz Guimarães, Celso Amorim Salim

**Affiliations:** Fundação Jorge Duprat Figueiredo de Segurança e Medicina do Trabalho—Fundacentro, São Paulo 30180-100, Brazil

**Keywords:** occupational exposure, public health, pilots, co-pilots, flight attendants

## Abstract

Addressing the field of health and safety at work, the primary objective of the present systematic review was to analyze the organizational risk factors for aircrew health according to professional category (flight attendants and pilots/co-pilots) and their consequences. The secondary objective was to identify the countries in which studies were carried out, focusing on the quality of content of the publications. The Medline/Pubmed, Cochrane, Web of Science, and Scopus databases were searched for eligible studies according to PRISMA statements. The risk of bias and the methodological quality of the studies were assessed using the Newcastle-Ottawa scale and Loney tools. Of the 3230 abstracts of articles screened, 36 studies met the inclusion criteria. Most of the research conducted on risk factors for the work organization of aircrew was carried out in the United States and the European Union and had moderate or low-quality methodology and evidence. However, the findings are homogeneous and allow the most prevalent organizational risk factors for the health of aircrew to be determined, namely, high work demand, long hours, and night work. Consequently, the most pervasive health problems were sleep disturbances, mental health disorders, musculoskeletal disorders, and fatigue. Thus, the regulation of the aircrew profession must prioritize measures that minimize these risk factors to promote better health and sleep for these professionals and, consequently, to provide excellent safety for workers and passengers.

## 1. Introduction

In Brazil, the demand for studies on the working conditions of aircrew and risks they are exposed to during work shifts has been growing in recent years. These demands stem from associations and organizations representing aviation professionals, as well as from airlines and government seeking to determine the impact on the social security system. Hence, these three dimensions intrinsically involve the regulatory framework governing the aviation industry and, consequently, related public policies.

In recent years, the civil aviation sector has grown significantly. According to the Brazilian Civil Aviation Agency (ANAC), in 2019, a record number of passengers (119.4 million) were carried on commercial flights in the country [1]. However, this high work demand, together with changes in flight schedules, stress, and psychological pressure, can cause fatigue, directly affecting the health of aircrew (Melo and Silvany Neto, 2012). This situation has given rise to a number of issues, such as instability, intense work pace, long working hours, irregular working hours, reduced pay, and loss of control over job activities [2,3].

Previous studies by Goode [4], Powell et al. [5], Marqueze et al. [6], Goffeng et al. [7], and Pellegrino and Marqueze [8] have highlighted that organizational aspects, such as long working hours, work demands, and schedules, numbered among the main risk factors for aircrew health. However, no study was found in the literature that summarized the findings on risk factors for these professionals. A study that synthesizes the main risk factors and their consequences for the aircrew’s health will be important to be able to think about health polices for these workers, as well as to synthesize what we already know about the subject and what still needs to be researched. Therefore, the primary objective of the present systematic review was to analyze the organizational risk factors for the health of aircrew according to professional category (flight attendants and pilots/co-pilots), as well as their consequences. The secondary objective was to identify the countries in which studies were carried out, focusing on the quality of content of publications.

## 2. Materials and Methods

A systematic review was performed according to the guidelines of the PRISMA checklist (Preferred Reporting Items for Systematic Reviews and Meta-analyses) [9] available in the Appendix A and registered on PROSPERO under number 240012. Given this was a review of the literature, approval by the Ethics Committee was not required.

The PICO strategy for non-clinical research was used to devise the research question (Table 1): what are the organizational risk factors for aircrew health? Pubmed, Cochrane, Web of Science, and Scopus databases were systematically searched from February to May 2021 to identify studies related to the research question. 

Keywords were defined according to medical subject headings (MeSH) and are presented in Table 2. The references of each study retrieved were manually investigated to identify additional eligible studies.

Potentially relevant studies were selected by two independent reviewers based on the following inclusion criteria: (1) studies in Portuguese, English, and Spanish only; (2) full studies; (3) observational (cross-sectional, cohort, case-control) and intervention studies only; (4) articles published after 1990; (5) studies involving men and women from any age group; (6) studies whose objective was to assess occupational risks, analyze their impact on aircrew health, and describe ways of mitigating these factors; (7) peer-reviewed studies; (8) studies on commercial aviation only. Exclusion criteria included clinical trials, ecological studies, reviews, qualitative studies, studies of retired aircrew, studies with objectives different from those of the present review (off-topic), abstracts, technical reports, oral communications, letters to the editors, and studies of flight simulation.

Article selection was performed independently, in a double-blind fashion, by two reviewers who screened all titles and abstracts. Subsequently, both reviewers read the full papers that met the inclusion criteria. In the case of disagreement regarding the eligibility of studies, a third researcher was consulted to reach a consensus.

The researchers created an electronic synthesis form to extract data from the papers reviewed. Data extraction covered the following variables: authors, publication year, study design, sample, study aim, main results, and study limitations declared by the authors. Microsoft Office Excel 2021^®^ was used to manage the selection of articles.

The risk of bias and methodological quality of studies involving human subjects were assessed using the Newcastle-Ottawa tools for cohort and case-control studies [10]; and Loney’s criteria [11] was used for cross-sectional studies. As described in the Newcastle-Ottawa scale, the methodological quality score of the cohort and case-control studies was calculated based on three components (range 0–9 points): (1) selection of groups (0–4); (2) quality of adjustment for confounders (0–2); and (3) ascertainment of exposure after outcome (0–3) [10]. Cross-sectional studies, in turn, were evaluated using the Loney criteria, (range 0–8 points), in which higher scores indicate superior methodological quality. This score is obtained from the eight questions making up the scale, and one or zero is assigned to each of the questions evaluated, according to the adequacy of the methods and presentation of the results [11].

## 3. Results

Application of the search strategy led to the retrieval of 3230 records, of which 1942 duplicates were removed. The remaining 1288 studies were screened based on title and abstract, of which 1223 were excluded for not meeting the inclusion criteria. Of the remaining 64 studies, 28 were excluded because they were not eligible, giving 36 studies for inclusion in the systematic review. Consensus was reached between the two reviewers on all articles selected (Figure 1).

### 3.1. Study Characteristics

The place of origin of the publications reviewed is shown in Figure 2. Most of the studies were carried out in the United States (22.2%), followed by Germany (11.1%) and Sweden (11.1%). Brazilian studies accounted for 8.3% of the publications selected.

The studies involving flight attendants were conducted between 1998 and 2019, 25% of which were published in 2019 (Table 3). The studies involving pilots were carried out between 1997 and 2020, with 30.4% published in the last five years (Table 4). Lastly, studies involving both flight attendants and pilots were conducted over the period 2002–2021, 70% in the last five years (Table 5). Overall, 36.1% of the studies reviewed were published during the period spanning from 2017 to 2021. 

The analysis of the results found in the studies involving flight attendants revealed that the main risk factors for work organization were high work demands (physical and psychological), harassment, and low job experience. The main consequences of these risk factors included sleep problems, cardiovascular diseases, mental health disorders, musculoskeletal disorders, injuries, and low work capacity (Table 3).

Lowden and Åkerstedt [12], in a prospective cohort study of cabin crew over a nine-day period, found that shorter rest time between work shifts (flights with quick layovers) led to decreased sleep time and efficiency. Sleep was not restorative, making it difficult to wake up and, consequently, increasing drowsiness during wakefulness. In the retrospective cohort study of Zeeb et al. [13], involving male cockpit crew members only, it was found that those who started working at age 30 or older had more than double the risk of cardiovascular mortality compared to professionals who entered the profession earlier. No risk gradient was observed for duration of employment (Table 3).

Ballard et al. [14], in a cross-sectional study of flight attendants, found that low job satisfaction and sexual harassment by passengers were risk factors for psychological distress. Lee et al. [15], also in a cross-sectional study involving flight attendants, identified both physical and mental work demands as risk factors for musculoskeletal disorders. Furthermore, in a cross-sectional study conducted by Castro et al. [16] investigating cabin crew members, the authors lack of humidity/moisture in the cabin, longer working experience, and being female were also risk factors for fatigue, poor sleep quality, and body aches. These factors, as well as their outcomes, can contribute to decreased work capacity (Table 3).

In a cross-sectional study by Hu et al. [17] of cabin crew, purser, and cabin manager, the authors confirmed that exhaustion from work, emotional distress, and insomnia were risk factors for decreased work ability. However, in a cross-sectional study of flight attendants conducted by Widyanti and Firdaus [18], a moderate mental workload was noted, regardless of flight duration. Lastly, Agampodi et al. [19], in a cross-sectional study of flight attendants, found that females (the predominant gender in this role), low weight (under 56 kg), and shorter work experience (<7 years) were risk factors for injuries on board (Table 3).

The most investigated topics among pilots were fatigue, drowsiness and alertness, which are frequent complaints among these professionals. In chronological order, the first study [20] with a prospective cohort design showed that long-haul flights, night flights, and short rest time between flights were associated with fatigue. In a cross-sectional study, Gander et al. [21] demonstrated that night flights, as well as short sleep duration, increased fatigue and decreased alertness levels led to poor eating habits, headaches, and backache among pilots. According to a cross-sectional study by Eriksen and Åkerstedt [22], night flights involve higher levels of sleepiness than morning flights for pilots (Table 4).

In a three-month prospective cohort study, Powell et al. [5] reported an association of long working hours and greater number of daily flights with a higher level of fatigue. Similarly, Roach et al. [23], in a prospective cohort, found that long working hours and sleep restriction increased fatigue and decreased level of attention among pilots. In a cross-sectional study carried out by Gander et al. [24], long working hours, with consequent sleep restriction on trans-meridional flights (crossing 7–9 time zones westwards), were also shown to affect pilots’ alertness level (Table 4).

In a cross-sectional study, Reis et al. [25] found that short-and medium-duration flights were factors associated with mental and physical fatigue among pilots. The authors explained this result as due to a large number of takeoffs and landings in shorter flights, events that demand the greatest attention and action of pilots. In a cross-sectional study by Gander et al. [26], the authors found that at top of descent (TOD—computed transition from cruise phase of flight to descent phase) and flights that lasted between 6:00 and 9:59 h led to most fatigue and impairment of pilots’ reaction time. Vejvoda et al. [27], in a cross-sectional study, also identified extended wakefulness and short-haul flights ending late at night as risk factors for fatigue (Table 4).

In a cross-sectional study, Van Drongelen et al. [28] showed that higher age, being an evening type, disturbance of work and personal-life balance, more need for recovery from work, lower perceived health, less physical activity, and moderate alcohol consumption were risk factors for fatigue. Lastly, Arsintescu et al. [29], also in a cross-sectional study, observed that higher workload, short sleep, greater number of daily flights, and short flights were factors associated with fatigue and impairment of pilots’ reaction time (Table 4).

According to Goode [4], pilots’ long working hours are risk factors for work accidents. Working days that last approximately 10–12 h increase the risk of accidents by 66%, and days of 13 h or more were associated with a 4.62 times greater risk. Another major problem among pilots is unintentional sleep during flights. In a cross-sectional study by Marqueze et al. [6], the prevalence of unintentional sleep was almost 60% among the pilots assessed. The authors found that longer monthly flying hours, frequent technical delays, greater need for recovery after work, work ability below optimal, insufficient sleep, and excessive sleepiness during waking hours were risks factors for unintentional sleep (Table 4).

Fatigue resulting from these risk factors can lead to other problems among pilots, as demonstrated by O’Hagan et al. [30]. In a cross-sectional study conducted by these authors, high work-related fatigue, frequent sleep disturbances, and unintentional naps during work were associated with depression and anxiety. Other factors were also associated with mental problems in this professional category. Feijó et al. [31], for instance, in a cross-sectional study, identified high work demand and sedentary lifestyle as risk factors for common mental disorders among pilots. The cross-sectional study by Demerouti et al. [32], also established that high demand and effort at work are risk factors for burnout in the same population. In addition to these factors, Wu et al. [33], in a cross-sectional study, found that the use of sleeping pills and harassment at work were associated with depression among pilots (Table 4).

Another frequent complaint among pilots is musculoskeletal disorder. In a cross-sectional study, Lawson et al. [34] observed that long flights are predictive factors for neck pain. Moreover, in a cross-sectional study, Runeson-Broberg et al. [35] identified having a high work demand, being female, having low social support, and suffering from psychosocial stress as risk factors for musculoskeletal symptoms. Albermann et al. [36], in a cross-sectional study, identified a high prevalence of chronic low back pain among pilots (82.7%) and a significant association with low back pain in those who had a total flight time of more than 600 h in the previous 12 months (Table 4).

Another problem studied that affects pilots was metabolic disorders. In a prospective cohort study, Zhao et al. [37] showed that pilots aged 31–50 years had a high prevalence of dyslipidemia compared to the general male population in China. De Souza Palmeira and Marqueze [38] observed a high prevalence of overweight among pilots (53.7%). Longer night work hours, difficulty relaxing after work, short sleep duration, chronic diseases, and a sedentary lifestyle were associated with excess weight. These problems, according to Bhat et al. [39], can lead to the development of other diseases. The authors found a positive correlation between overweight and hypertension among pilots, especially in individuals aged 26–35 years (Table 4).

Based on the results obtained in the studies of pilots only, the main organizational risk factors for the health of aircrew were long working hours, night work, little rest time between workdays, sleep restriction, and long waking hours due to working hours, high number of daily flights, low social support, organizational stress, frequent technical delays, lack of work and personal life balance, high work demand, and long-haul flights. The main health outcomes of these factors were physical and mental fatigue, lower alertness, poor eating habits, headaches, muscle pain, injuries, mental disorders, metabolic problems, unintentional sleep, and excessive sleepiness (Table 4).

In studies of flight attendants and pilots, other health issues were investigated besides fatigue. Ballard et al. [40], in a retrospective cohort study evaluating the association of causes of death (malignant neoplasms, non-cancerous, non-injurious, and external causes) with job position and gender, found that males with longer working hours were at higher risk of leukemia, while flight attendants (female) were at higher risk of suicide. In the cross-sectional study of Omholt et al. [41], flight attendants had a higher prevalence of musculoskeletal symptoms, as well as psychological, gastrointestinal, and allergy complaints, compared to pilots. In total, 20% reported high stress levels, which were associated with these symptoms and complaints. Regarding restless legs syndrome (RLS), Düz et al. [42], in a cross-sectional study, found that the prevalence of RLS among pilots and flight attendants was similar to that of the general population, and that flying at high altitudes was not a risk factor for the syndrome (Table 5).

The studies by Goffeng et al. [43] and Åkerstedt et al. [44] evaluated both pilots and flight attendants. The first of these investigations, a seven-day prospective cohort study confirmed higher number of consecutive workdays (≥days), short sleep, less rest time between flights, and high workload as risk factors for greater cardiovascular effort among pilots and flight attendants. More specifically, Åkerstedt et al. [44], in a 14-day prospective cohort of flight attendants, found that long working hours, night work, short sleep, and very early flights were risk factors for fatigue in this group (Table 5).

In summary, these studies found that the main work organization risk factors for these professionals were longer work experience, consecutive working days, sleep restriction due to working hours, high work demand, long working hours, and very early flights. The main outcomes of these factors were the development of cardiovascular diseases, mental disorders, and fatigue (Table 5).

**Table 5 ijerph-20-03401-t005:** Characteristics and main results of studies involving both flight attendants and airline pilots.

Authors (Year)	Study Design	Sample	Aim	Main Results	Limitations Declared by the Authors
Pilots and Flight Attendants					
Ballard TJ, Lagorio S, De Santis M, De Angelis G, Santaquilani M, Caldora M, Verdecchia A (2002)	Cohort retrospective	An amount of 3022 male (28.6 years, 20.4–61.2 yrs) and six female cockpit crew members (29.9 years, 23.6–33.5 yrs); 3418 male (25.6 years, 18.8–60.1 yrs) and 3428 female cabin attendants (22.6 years, 18.9–59.7 yrs)	To evaluate the association between causes of death (malignant neoplasms, non-cancer, non-injury causes and external causes) according to position level and sex.	This study demonstrated reduced risks among Italian male cockpit crew members for all-causes and all-sites cancer mortality with respect to the reference population. Mortality from all cancers was less than expected for all categories (SMRs of 0.58 for male cockpit crew, 0.67 for male cabin attendants, and 0.90 for female cabin attendants). Among male flight personnel, the SMR for leukemia was somewhat elevated (SMR 1.73; 95% CI: 0.75–3.41) based on eight deaths, with a positive trend by length of employment (*p* = 0.046). Additionally, an excess of death by suicide was seen among female cabin attendants (SMR 3.38; 95% CI: 1.24–7.35).	The cohort was relatively young, with few deaths, resulting in imprecise risk estimates. This reduced the ability to identify associations between occupational exposures, including exposure to cosmic radiation, and mortality from cancer, if such an association exists. For persons exposed to 100 mSv of cosmic radiation over a lifetime career of flying at high altitudes, the overall excess lifetime cancer mortality risk due to cosmic radiation is very low, estimated at 0.5%, compared with a lifetime cancer mortality risk of 25% from all other possible causes. For breast cancer, the presumed relative risk is 1.04 for occupational exposure to cosmic radiation, a rate that is extremely difficult to detect in an epidemiologic study of moderate dimension such as this one. A second limitation of this study is related to the use of length of employment as a surrogate of absorbed radiation dose. To address the validity of various measures of exposure to cosmic radiation, a team of European researchers conducted a correlation study of four methods for estimating absorbed cosmic radiation, using flight data of German pilots. It was found that length of employment, although the least specific of the measures, did correlate somewhat with more precise methods of estimating dose.
Omholt ML, Tveito TH, Ihlebæk C (2017)	Cross-sectional study	An amount of 843 aircrew members: cockpit crew 28% (4% female, 46% 41–50 years; cabin crew 17% (75% female, 34% 31–40 years)	To investigate the relationships between work-related stress, self-efficacy and subjective health complaints (SHCs) in commercial aircrew in Norway and to explore differences between cockpit and cabin crew.	Tiredness, sleep problems, bloating, low back pain, headaches, and neck pain were the most prevalent SHCs. Cabin crew reported significantly higher numbers, prevalence and mean values for all SHCs compared with cockpit crew (*p* < 0.05). In total, 20% reported high stress levels. High levels of work-related stress were significantly associated with all SHC factors in both groups. Self-efficacy partly moderated the relationship between stress and psychological complaints in both cockpit and cabin crew, and for musculoskeletal complaints in cockpit crew. The model explained 23 and 32% of the variance in psychological complaints for cockpit and cabin crew, respectively. High levels of work-related stress were significantly associated with the level of musculoskeletal, psychological, gastrointestinal and allergic complaints.	One weakness was the low response rate, and as no data were available on how many aircrew were actually reached and invited and no information on the non-responders, possible selection bias cannot be ruled out. The prevalence values might therefore not exactly represent the general population of Norwegian aircrew members since an over-representation of healthy subjects in health surveys has been reported. Furthermore, there was no information on the frequency or duration of flights or on work schedules and had to rely on the information aircrew members gave on whether they currently worked on domestic, Scandinavian, European and/or intercontinental flights. Lastly, the cross-sectional design of the study precluded the drawing of any conclusions about any causal relationships.
Düz OA, Yilmaz NH, Olmuscelik O (2019)	Cross-sectional study	A total of 301 actively flying Turkish aircrew (192 pilots, 109 cabin crew)—37.4 yrs (24–63 yrs), 22.6% female—and 272 age- and sex-matched healthy subjects—39.0 yrs (23–63 yrs), 26.1% female	To explore the frequency of restless legs syndrome (RLS) in aircrew.	The impact of being at a high altitude on RLS is controversial. The RLS frequency in aircrew was 6.7%, which is similar to that of the normal population. It is considered that this similarity is due to modern technology which regulates and adjusts oxygen saturation and air pressure inside the aircraft throughout the flight. We can conclude flying at high altitude was not a risk factor for RLS.	One limitation of this study is daily sleep duration was measured subjectively by self-reporting of the participants; objectively measured sleep gives more information about the relationship between RLS and being an aircrew member.
Goffeng EM, Nordby KC, Tarvainen M, Järvelin-Pasanen S, Wagstaff A, Skare Ø, Lie JÁ (2019)	Cohort prospective (7 days)	An amount of 17 pilots (15 men; mean age 52 yrs, SD 12.3 yrs) and 41 cabin crewmembers (six men; mean age 40 yrs, SD 7.4 yrs)	To evaluate changes in heart rate variability (HRV) during an actual flight duty period and sleep, and with respect to work characteristics and breaks.	The results indicate higher levels of cardiovascular strain on the 4th compared to the 1st workday, most prominent among cabin crewmembers. In this group, there were indications of decreased cardiovascular strain by increasing duration of sleep, demonstrated by increased root mean square of successive differences (RMSSD) (B = 2.7, 95% CI 1.6, 3.8) and standard deviation of the normal beat-to-beat differences (SDNN) (B = 4.4, 95% CI 3.0, 5.7), and decreased low and high frequency ratio (LF/HF) (B = 20.2, 95% CI, 20.4, 20.01). Similarly, longer duration of breaks was associated with lower cardiovascular strain, indicated by increased RMSSD (B = 0.1, 95% CI 0.03, 0.1) and SDNN (B = 0.1, 95% CI 0.1, 0.1). Among pilots, increased LF/HF indicated higher cardiovascular strain in those who often or always reported of high workload (B = 4.3, 95% CI 2.3, 6.3; and B = 7.3, 95% CI 3.2, 11.4, respectively).	One limitation of the study was the small sample size, particularly the small pilot group, which reduced statistical power and thus the capacity to detect differences and trends observed at a borderline statistical significance. This was partly modified by the repeated-measurement design. Furthermore, the study population was not a random sample, which may have resulted in selection bias, and decreased the generalizability of the results. The skewed gender distribution within the group of cabin crewmembers and pilots in most airlines. Finally, although though the number of work hours was similar among all subjects, the exact times for check-in and check-out for duty varied, which may also have influenced the results.
Åkerstedt T, Klemets T, Karlsson D, Häbel H, Widman L, Sallinen M (2021)	Cohort prospective (7–14 days)	An amount of 106 aircrew, age (38.3 yrs, SD 8.6 yrs), 76% male, 84.2% flight deck and 7.2% cabin crew	To investigate the associations of schedule characteristics with fatigue and amount of sleep in the acute 24-h window, and as cumulative effects across the seven-day work period.	For the 24-h window (acute), all variables entered singly were significantly associated with fatigue. Duty time, block time, sectors, time from start and non-day duty types were all associated with increased fatigue, whereas amount of sleep was associated with decreased fatigue. When all variables were entered at the same time, sleep, non- daytime duties and duty time retained their significant regression. Block time, hours since start and number of sectors lost their explanatory power due to the influence of duty type and sleep. For the seven-day work period, the results indicate that the accumulation of early and very early duties was associated with increased fatigue, whereas accumulation of sleep was associated with decreasing fatigue. When accumulated sleep was entered into the regression, accumulated duty time, number of sectors, and block time became associated with increasing fatigue, and early duties lost their significant association. The results suggest that sleep, duty time, and early starts are important predictors of fatigue in the 24-h window and that the number of very early starts and short sleep have cumulative effects on fatigue across a seven-day work period.	Among the limitations of the present study is that the dependent variable was based on self-report. Yet, there are no well established objective measures of fatigue usable in real-life work situations. The psychomotor vigilance test is a well established fatigue measure under controlled conditions (Lim and Dinges, 2008) [45], but it has never been validated against real work performance during a work shift and would take too long to carry out in a study with many (short) sectors. Another limitation is that some participants did not provide information on age and gender, probably due to concerns of anonymity. Another limitation is that no data were collected on bedtimes and times of rising, which would have made it possible to evaluate effects of time awake. A weakness is also that the study did not attract sufficient numbers of cabin crew to make a proper evaluation of that group. Finally, the results are only generalizable to daytime operations, albeit with a wide span of early and late flights.

Notably, none of the studies included in this review researched ways of mitigating the risk factors investigated. The study authors only suggested measures to minimize health risks, including a reduction of consecutive workdays, especially night work, longer rest time between shifts, particularly after long-haul flights crossing time zones, provision of a suitable rest area on planes, optimizing the work by support team to shorten flight turnaround times, avoiding work schedules with very early starts, having greater social support at work, and prioritizing rest days on weekends.

### 3.2. Quality Assessment

The results of the quality assessment of the study designs according to the Newcastle-Ottawa or Loney criteria are given in Table 6. Out of the 36 articles reviewed, eight (22.2%) involved flight attendants only, 23 (63.9%) pilots only, and five evaluated both flight attendants and pilots (13.9%). Regarding the eight studies of flight attendants only, two (25%) were cohort studies, and six were cross-sectional (75%). Only one of the cross-sectional studies was rated with 7 points, i.e., of ‘high methodological quality’. Two studies were rated with five points, one with four points, and two studies were attributed a rating of 3 points, indicating that studies were predominantly of low methodological quality.

Regarding the studies of pilots only (*n* = 23), most were cross-sectional (*n* = 19, 82.6%). Of this total, four were rated with 6 points, indicating better quality. Eleven studies had a rating of 5, and two were attributed four points, indicating moderate quality. The remaining two studies scored only 3 points, i.e., were of low methodological quality. Only four publications were prospective cohort studies (17.4%), while none had a high level of evidence. Three of the studies had a moderate level of evidence, comprising two with 5 points and one with 4 points, while one had a low level of evidence (3 points) (Table 6).

For the three studies involving both flight attendants and pilots, most were cohort studies (two prospective and one retrospective—60%). Of these studies, two had moderate evidence (one with four points and another with five points), and only one study had low evidence. Of the two cross-sectional studies (40%), one scored five points, indicating moderate quality, and the other scored three points, indicating low quality (Table 6).

## 4. Discussion

The most prevalent organizational health risk factors faced by flight attendants and pilots were long hours, high work demand, and night work. These same risk factors are often found in other work activities also involving irregular hours and night shifts. An estimated 30% of the adult population are engaged in shiftwork, including night work [46]. According to the Working Time Society consensus statements, shift and night work are risk factors for workers’ health, as circadian desynchronization, sleep restriction, and social misalignment can cause health problems (directly or indirectly) as a result of exposure tonight work [47]. Shift and night work lead to acute and chronic disturbances in sleep and alertness, increased risk of fatigue-related accidents and incidents, cardiovascular problems, metabolic disorders, musculoskeletal disorders, increased risk of developing cancer (prostate, colorectal and breast), unhealthy behavioral changes, such as those related to timing of meals and low physical activity, and gastrointestinal and digestive disorders [46,47,48,49]. Moreover, mental health problems are common in shift workers. According to Boivin et al. [46], these conditions develop due to long-term exposure to work schedules, leading to the disruption of the sleep/wake cycle and circadian desynchronization, significant risk factors for mental health.

In a systematic analysis carried out jointly by the World Health Organization (WHO) and the International Labor Organization (ILO), individuals who work long hours have a higher risk of developing ischemic heart disease and stroke [50]. Other systematic reviews have also revealed that long working hours are associated with burnout syndrome [51], decline cognitive capacity [52], depression [53,54], increased alcohol consumption [55], mental health disorders [56], and adverse pregnancy outcomes (preterm birth, chance of miscarriage, low birth weight) [57]. High work demands related to working hours and shift and night-work are also associated with problems such as burnout and anxiety [58], musculoskeletal disorders [59], sleep disorders [60], emotional exhaustion [61], and sedentary lifestyle [62].

In the case of aircrew, in addition to hours spent in flight, commuting time, especially for professionals living far from their designated place of work or staying in hotels, should be taken into account. Hotels located far from airports will mean longer commutes to work. Furthermore, the authors note that, even when at work base, most airports are far from cities and require a long commuting time. Moreover, it is necessary to account for the time needed to carry out briefings and administrative activities, as well as the time engaged at the end of the workday, which ends not when the engines are shut off but when aircrew are at home to rest. Technical delays are also a factor that can extend work hours of aircrew and should thus be taken in account [6].

As a result of these risk factors, the most prevalent health problems reported in the studies analyzed were related to sleep, mental health disorders, musculoskeletal disorders, impaired cognition and performance, and fatigue. Sleepiness and impaired cognition and performance are common among airline pilots [63,64,65]. In a study by Bostock and Steptoe [66] of airline pilots, based on an analysis of salivary cortisol, the authors found a decrease in wake-up response time on shifts that started either very early or very late, directly impacting sleepiness and impaired cognition and performance. According to Kecklund and Axelsson [67] and Fischer et al. [68], shift work and its deleterious effects on workers’ sleep and concentration impair alertness levels and increase the risk of fatigue-related accidents.

As reported by Wingelaar-Jagt et al. [69], fatigue is related to several factors, including sleep debt, long wakefulness, circadian desynchronization and high workload, has several deleterious effects on health, and decreases aircrew performance during flight. Problems associated with fatigue in aircrew include decreased alertness, mental health problems, excessive sleepiness, sleep disturbances, and increased cardiovascular effort [8,43,70,71]. Although regulations limit flight time, it is noteworthy that fatigue cannot be mitigated entirely and the best way to avoid it is to obtain adequate sleep at night for recovery [69].

Wingelaar-Jagt et al. [69] identified some factors in aviation that contribute to fatigue, namely, long-haul flights, mainly as a result of jet lag, night flights, and circadian rhythm disturbances; short-haul flights, because of the high work demand and sleep loss for having to wake up very early to work or sleep late due to late flights; international flights; problems with work schedules, such as many consecutive nights of work, lack of a suitable rest area on the aircraft, and flight schedules, especially those starting very early or ending late at night; high number of flights; and long working hours, while taking into account, in addition to flight time, the time dedicated to briefings, administrative activities, etc.

Regarding the quality of the studies analyzed in the present review, most were cross-sectional (75%). In cross-sectional designs, a cause-and-effect relationship between risk factors and outcomes cannot be established, only indicating the existence of any associations between the variables studied [72]. However, these studies contribute significantly to the understanding of which organizational factors are associated with the health problems of aircrew, as well as helping identify the most frequent issues.

These data are important for designing and planning new studies, specifically longitudinal studies to elucidate the impact of work organization on the health of these workers and contribute to the management and care of these professionals. It is also essential to carry out dose-response studies to assess biological and behavioral factors that may impact individual susceptibility to the work schedules to which flight attendants and pilots are exposed, in addition to identifying mediating factors between risk factors and health outcomes. Social, economic, and legal aspects should be simultaneously considered and evaluated [45]. On the other hand, it is important to emphasize that most of the analyzed studies (86.1%) were classified as having moderate or low methodological and evidence quality, pointing to the need for more robust studies with fewer biases.

Finally, when considering the places where the studies were carried out, according to the Science and Engineering Indicators report [73], the United States and the European Union are the places that publish the most scientific studies in the worldwide, especially in the area of health. This same trend was also found in the present review, which analyzed studies conducted mainly in the United States, followed by Germany and Sweden.

This review contributes to the growing body of evidence suggesting that the work of aircrew has relevant risk factors which need to be better understood. Further studies, especially those with longitudinal designs, should be carried out to provide more robust evidence on organizational risk factors to the health of this group of professionals and to assess the actions and measures which could be taken to mitigate these risks.

## 5. Conclusions

In short, most of the research conducted on risk factors for the work organization of aircrew was carried out in the United States and the European Union and had moderate or low-quality methodology and evidence. The findings are homogeneous and allow the most prevalent organizational risk factors for the health of aircrew to be determined, namely, high work demand, long hours, and night work. As a result of these factors, the most prevalent health problems in both professional categories were those related to sleep, mental disorders, musculoskeletal disorders, and fatigue.

Our findings have interesting implications for gaining a better understanding of the main risk factors of aircrew health since this job is significant for society and guides future discussion about the regulation of the aircrew profession. The present review makes it clear that further studies with this professional category need to be carried out, especially longitudinal studies, to understand the causal relationships between occupational risk factors to health.

## Figures and Tables

**Figure 1 ijerph-20-03401-f001:**
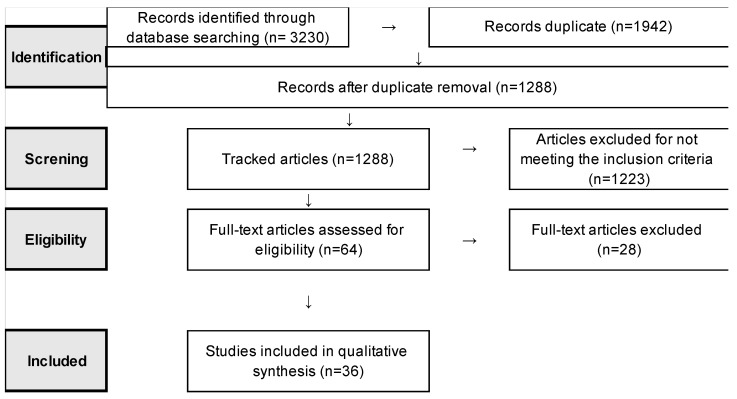
Search and selection process according to PRISMA statement.

**Figure 2 ijerph-20-03401-f002:**
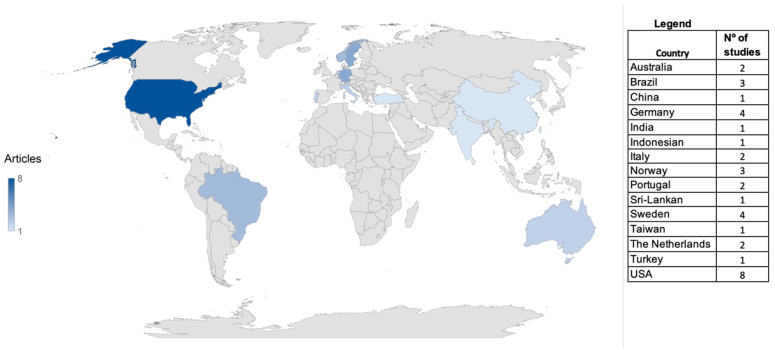
Countries of publication of studies reviewed.

**Table 1 ijerph-20-03401-t001:** PICO strategy for devising the research question.

Criteria	Definition
Population	Commercial aircrew
Interest	Work organization
Context	Occupational risk factor

**Table 2 ijerph-20-03401-t002:** Search terms on electronic databases.

Outcomes	“Occupational Risks” or “Risk Factors” or “Work hours” or “Workload” or “Shift Work Schedule” or “Night Shift Work”
Population	“Aircrew” or “Flight attendants” or “Air Pilots” or “Aviators” or “Co-Pilot”
Limit	“Full-text available” and “Peer-reviewed papers”

**Table 3 ijerph-20-03401-t003:** Characteristics and main results of studies of flight attendants.

Authors (Year)	Study Design	Sample	Aim	Main Results	Limitations Declared by the Authors
Flight Attendants					
Lowden A, Åkerstedt T (1998)	Cohort prospective (9 days)	An amount of 42 cabin crew, 48% women, 42 yrs (SD 9 yrs)	To describe the spontaneous sleep/wake pattern in connection with a westward (Stockholm to Los Angeles) transmeridian flight (−9 h) and short layover (50 h)	During the outbound day the wake span was 21.7 h and 90% of the aircrew adopted local bed times on layover. The readaptation to normal sleep/wake patterns were rapid on the return. Napping was common (93%), especially on-board and before the return. Sleep efficiency dropped below 90% during layover, being felt to be too short and disturbed by awakenings, and gradually returned to normal across four recovery days. Recovery sleep was characterized by difficulties waking up and feelings of not being refreshed from sleep. Sleepiness symptoms increased during layover and gradually decreased across recovery days, still being elevated on day 4. Westward flights are associated with extended wake spans during layover, increased sleepiness, and slow recovery on return home. Strategic sleeping may counteract the effect somewhat, but individual differences are few.	None declared by the authors.
Zeeb H, Langner I, Blettner M (2003)	Cohort retrospective	An amount of 6061 male cockpit crew	To investigate the cardiovascular mortality of German cockpit crew	Among the total of 255 deaths recorded in the cohort between 1960 and 1997 (SMR = 0.5; 95 confidence interval 0.4–0.5), there were a total of 58 cardiovascular deaths (ICD-9 390–429), yielding an SMR of 0.5 (95% CI 0.3–0.6). Overall mortality from cardiovascular causes among cockpit crew was reduced. For acute myocardial infarction the SMR was 0.4 (95% CI 0.3–0.7). Cockpit crew taking up employment at age 30 or later had a more than twofold cardiovascular mortality risk compared with those beginning employment earlier, but there was no risk gradient with duration of employment. Overall, cockpit crew has a relatively low cardiovascular mortality to which a low smoking prevalence and an early detection of cardiovascular health problems are likely to contribute. Cockpit crews employed before age 30 have the lowest cardiovascular mortality risk.	Limitations of the retrospective study approach include the reliance on official death certificates and the absence of information on possible cardiovascular risk factors in the cohort. The cohort among cockpit crew in Germany showed a low mortality from cardiovascular causes which persisted over time and among older and retired pilots. An early entry into employment was associated with particularly low cardiovascular mortality. Several factors, among them the good health status at hire and an early detection and intervention in case of abnormalities, are likely to contribute to the observed low cardiovascular mortality.
Ballard TJ, Romito P, Lauria L, Vigiliano V, Caldora M, Mazzanti C, Verdecchia A (2006)	Cross-sectional study	An amount of 1955 flight attendants: 48% in service (mean age 37.1 years, 25–58 years) and 52% not in service (mean age 50.2 years, 27–75 years)	To investigate associations of work-related risk factors with self-perceived health as less than “good” and psychological distress among Italian women flight attendants	The results of this cross-sectional study of a cohort of current and former Italian women flight attendants showed that current flight attendants perceived their health as only fair or poor and reported psychological distress more frequently than former flight attendants (47.0% v 40.6%, respectively, for fair to poor health and 17.2% v 12.7%, respectively, for psychological distress). Among current flight attendants, these outcomes were related to low job satisfaction and to experiencing sexual harassment by passengers, the latter being especially strong for perceived health.	Potential limitations to this study include the cross-sectional nature, which precludes identifying causal risk factors for perceived fair to poor health or psychological distress. Another limitation is that the response rate was greater among current compared with former flight attendants, which may have affected the comparisons made between the two groups. However, the large response rate of currently working women provided a greater possibility that the high prevalence of perceived poor health and distress and their association with low job satisfaction, family conflicts, and experiences with sexual harassment reflect the realities of this type of work.
Lee H, Wilbur J, Kim MJ, Miller AM (2008)	Cross-sectional study	An amount of 164 flight attendants was 54 years (SD = 6.2), ranging from 32 to 68 years	To examine the relationships between work-related psychosocial factors and lower-back work-related musculoskeletal disorders among long-haul international female flight attendants	The prevalence of lower-back WMSDs (60.4%) was very high in this sample of U.S. flight attendants. The physical load of job tasks performed by flight attendants working long-haul international flights are related to lower-back WMSD. The flight attendants with lower back work-related musculoskeletal disorders, compared with those without lower-back work-related musculoskeletal disorders, had higher perceived psychological job demands, job insecurity, and physical load. After controlling for physical load and personal factors, high job insecurity significantly increased the risk for lower-back work-related musculoskeletal disorders.	The sampling frame was limited to flight attendants who were scheduled for an international flight during one month of the year. In addition, the response rate was lower than estimated based on a conservative response rate of 50% and additional 25% by two more follow-up mailings.
Agampodi SB, Dharmaratne SD, Agampodi TC (2009)	Cross-sectional study	An amount of 98 (30.4%) male and 224 (69.6%) female flight attendants, mean age 31 years (SD 8 yrs)	To estimate the incidence of onboard injury among Sri Lankan flight attendants and to describe the determinants of onboard injury	A total of 100 onboard falls, slips, or trips in the previous six months were reported by 52 (16.1%) respondents. Of the total sample, 128 (39.8%) cabin crew members reported an injury in the six months preceding the study. This represents a total injury incidence of 795 per 1000 person per year. The leading causes of injury was pulling, pushing, or lifting (60.2%). The commonest type of injuries were strains and sprains (52.3%). Turbulence-related injuries were reported by 38 (29.7%) flight attendants. The upper limbs (44.5%) and the back (32%) were the commonest sites affected. After controlling for other factors, female flight attendants had 2.9 times higher risk (95% CI 1.2–7.2) of sustaining and injury than males. Irrespective of sex, body weight less than 56 kg (OR 2.9, 95% CI 1.4–5.8) and less than seven years of on board experience (OR 10.5, 95% CI 3.6–31.0) were associated with higher risk of injury.	The sample was not a proper random sample, consisted of batches of flight attendants recruited in the same time period in different years. Differences in training procedures could affect the outcome of the present study. The second limitation was recall bias. Reporting of injury is dependent on recall, which depend on individual characteristics, severity of injuries, and impact of the injury on the person.
Castro M, Carvalhais J, Teles J (2015)	Cross-sectional study	An amount of 73 cabin crewmembers (representing 61.9% of the population), 39 females (53.4%), aged between 20 and 37 years, with an average age of 27.68 (±4.27 years)	To analyze: what are the requirements of the cabin crew work; whether the schedules being observed, and effective resting timeouts are triggering factors of fatigue; and the existence of fatigue symptoms in the cabin crew	The data indicate the presence of fatigue and corresponding health symptoms among the airline cabin crew, despite the favorable sample characteristics: relatively young, with low seniority (activity exposure), free of commitments and with healthy lifestyles. Senior workers are significantly more affected by the lack of humidity in the cabin, and they also have more problems in sleep and stomach. The results seem to indicate that higher exposure to this activity (increased seniority), together with increasing age, could lead to more fatigue and health problems, namely poor sleep quality. Regarding the low seniority found in this study, we can speculate that perhaps there is a kind of natural selection, making that people with more difficulties leave the company or the activity of flight attendant. Furthermore, women tend to be more affected by several fatigue factors than men and also, they have more complaints about body pain and some other fatigue symptoms.	None declared by the authors.
Widyanti A, Firdaus M (2019)	Cross-sectional study	An amount of 201 flight attendants (mean age = 24.6 years, SD = 3.8 years, all female)	To examine the mental workload of flight attendants, and it possible relation to the flight duration	The average mental workload of the flight attendants regardless the flight duration, measured using the NASA-TLX, is 76.08 (SD = 12.66) out of 100. The score indicates a medium level of mental workload.	This study has several limitations worth noting. First, caution is called for in generalizing the findings. The sample of the Indonesian flight attendants is only limited for one airline. Therefore, generalizability is limited. Second, the participants of this study were limited to female flight attendants due to the nature of the work of flight attendants which is dominated by female workers. Further research involves male flight attendants may enrich the result and the analysis. Third, analyze of the mental workload based on flight duration is designed as between subject study due to difficulty to match every flight attendants to every possible flight duration. Further research with a design of within subject comparison will enrich the analysis.
Hu CJ, Hong RM, Gwo-Liang Yeh GL, Hsieh IC (2019)	Cross-sectional study	An amount of 412 (87.3%) Women and 60 (12.7%) Men; 379 (80.3%) Cabin crew, 36 (7.6%) Purser and 57 (12.1%) Cabin manager; 306 (64.8%) < 45 yr and 166 (35.2%) ≧ 45 yr	To explore the current status and factors affecting the work ability of flight attendants	The average WAI score in the study was 39.9 6 3.8 and 82.4% of participants were classified as work ability of ‘good or above.’ Male sex, good eating habits, and the job title of cabin manager were most strongly associated with high WAI scores, while severe work-related burnout and severe emotional distress were associated with low WAI scores. Overall, insomnia had the most negative impact on flight attendants’ work ability.	None declared by the authors.

**Table 4 ijerph-20-03401-t004:** Characteristics and main results of studies of airline pilots.

Authors (Year)	Study Design	Sample	Aim	Main Results	Limitations Declared by the Authors
Pilots					
Samel A, Wegmann HM, Vejvoda M (1997)	Cohort prospective	An amount of 50 male pilots (25 captains and 25 flight officers) volunteered to participate in the three study phases. All of them had >two years of experience on the aircraft B 767–300. The average age was 35.0 years (SD = 8.4 years), ranging from 25 to 56 years	To evaluate the duty schedules on demands on mental and physiological capacity.	In flight ratings of task load showed low perceived exertion during the Atlantic flights and were moderate during the north–south transitions. Fatigue ratings increased with progressing flight duration. Towards the end of long U.S.-west coast flights performed at daytime, and in all night flights, fatigue was elevated compared to the ‘baseline’ ratings collected during the day-time DUS-ATL flights. Fatigue was rated as being ‘critical’ by several pilots, particularly during the return flight SEZ-FRA when fatigue was severely pronounced. At least 48 h of rest (if not more, to be on the safe side) are necessary for recovery from sleep deprivation after rotations as described in this report. In the case of transatlantic flights, the adaptation of the circadian system to LT after time-zone changes adds a further dimension to the problem of adequate rest.	None declared by the authors.
Gander PH, Gregory KB, Miller DL, Graeber RC, Connell LJ, Rosekind MR (1998)	Cross-sectional study	An amount of 25 flight crewmembers (10 captains, 7 first officers, 8 flight engineers), all men, 52.7 yr (SD 5.0 yr)	To monitor flight crewmembers before, during, and after 4–9 d commercial long-haul trips crossing up to 8 time zones per 24 h.	Greater sleep loss was associated with nighttime flights than with daytime flights. The organization of layover sleep depended on prior flight direction, local time, and the circadian cycle. The circadian temperature rhythm did not synchronize to the erratic environmental time cues. Consequently, the circadian low point in alertness and performance sometimes occurred in flight. On trip days, by comparison with pre-trip, crewmembers reported higher fatigue and lower activation, drank more caffeine, ate more snacks and fewer meals, and there were marked increases in reports of headaches, congested nose, and back pain.	None declared by the authors.
Goode JH (2003)	Cross-sectional study/Data base	Captain’s	To demonstrate an empirical relationship between pilot schedules and aviation accidents.	The proportion of accidents associated with pilots having longer duty periods is higher than the proportion of longer duty periods for all pilots. For 10–12 h of duty time, the proportion of accident pilots with this length of duty period is 1.66 times as large as for all pilots. For pilots with 13 or more hours of duty, the proportion of accident pilot duty periods is 5.62.	For the analysis, one month of flight activity data was compared with 20 years of accident data arguing that the distributions of these data sets should be the same if length of duty has no impact on accidents. However, one may wish to be cautious of the empirical findings based on this assumption. If these distributions were not expected to be the same in the first place, then the findings are problematic. With this potential weakness in mind, the findings do strongly suggest that there is an increased probability of an accident as duty time increases, and therefore more stringent limitations on pilot duty time may be appropriate.
Eriksen CA, Åkerstedt T (2006)	Cross-sectional study	Seven morning-crew pilots and seven evening-crew pilots. The average age of the morning flight pilots was 50.0 + 3.9 yrs (mean + SE). The average age of the evening flight pilots was 48.1 + 2.5 yrs. All male	To make a first attempt to compare sleepiness and sleep on westward morning and evening flights, as such a comparison might have a bearing on operational safety.	The results show that total sleep times did not differ between morning and evening flight crews, despite the fact that morning crews had to arise earlier before the flight and evening crews went to bed significantly later during the layover. Apparently, the pilots compensated for the effects of the flight by going to bed earlier, and sleeping longer in the morning, respectively. The total sleep time over the days varied, however, with shorter periods of sleep the night before the outward- bound flight and the greatest amount of sleep during the night after returning home, for both morning and evening flights. There was no difference between the morning and evening flights with regard to total sleep time, bedtimes, or rising times over the course of the study. Actigraphy showed no differences between morning and evening flights, which seems to indicate that the need for recovery was not affected by working during the daytime or night. Evening flights involve higher levels of sleepiness than morning flights, presumably because of the close proximity in time to the circadian trough of alertness.	None declared by the authors.
Powell DM, Spencer MB, Holland D, Broadbent E, Petrie KJ (2007)	Cohort prospective (3 months)	There were 1466 questionnaires returned, giving a response rate of 72% of total duty periods, and these were further reduced to 1370 (67%) after the application of the exclusion criteria. Most duties originated in Auckland (39%), followed by Christchurch (32%), Wellington (21%), and Dunedin (8%)	To investigate how length of duty, number of sectors, time of day, and departure airport affect fatigue levels in short-haul operations.	The most important influences on fatigue were the number of sectors and duty length. These were associated with fatigue in a linear fashion. Time of day had a weaker influence, with lower levels at midday and increased fatigue later in the day. Fatigue was also higher during duties originating from an airport where pilots needed to position the night before and spend the night in a hotel. Data from the study enabled the quantification of fatigue at this critical phase of flight in duties lasting between 2 and 12 h and finishing between 08:00 and 24:00.	There are a number of methodological issues relevant to the interpretation of the results. Firstly, fatigue was measured using subjective pilot ratings, rather than using objective indices, such as re- action time tasks. However, the Samn-Perelli scores have been shown to follow similar trends to objective measures throughout a duty period (8). Secondly, all the data were collected on domestic sectors of a similar length with no overnight duties and no time zone changes, so they may not be directly applicable to different operations. It is also likely that this lack of over- night duties attenuated the time of day effect as highest levels of fatigue tend to occur in the early morning.
Roach GD, Petrilli RM, Dawson D, Lamond N (2012)	Cohort prospective	An amount of 19 male pilots (10 Captains—mean (±SD) age of 53.9 (±1.9) yrs, 9 First Officers—mean age of 45.6 (±6.4) yrs)	To examine the impact of layover length on the sleep, subjective fatigue levels, and capacity to sustain attention of long-haul pilots.	The pilots in this study obtained substantially more sleep/d during their layover than they did during their pre-trip days off. Although the interaction was not statistically significant, it seems apparent that the effect of the layover on sleep was mainly due to the group of pilots that had a short layover. Indeed, these pilots obtained a mean of 8.6 h of sleep/d during the layover, compared to 7.2 h of sleep/d during pre-trip days off. As with most long-haul flights, the outbound flight in this study involved a very long duty day (~16 h). Thus, not surprisingly, pilots had substantially higher fatigue levels at the end of the flight than at the start, as indicated by their subjective ratings and their capacity to sustain attention. The need to recover from the sleep debt and fatigue accumulated during the outbound flight could account, at least in part, for the fact that pilots obtained more sleep during layover days than they obtained on days off at home prior to the trip. The results of this study indicate that a short layover during a long-haul trip does not substantially disrupt pilots’ sleep, but it may result in elevated levels of fatigue during and after the trip. If short layovers are used, pilots should have a minimum of 4 d off to recover prior to their next long-haul trip.	First, the number of participants was limited by time and cost restraints. If this resulted in the study being underpowered, then the likelihood of committing a type II error would have been elevated. Second, participants were not randomly allocated to the two layover groups by the authors, but instead they were assigned to a long-haul trip that contained either a short or a long layover in an optimization process conducted by the employer airline. If there was some bias in the assignment of trips—a possibility that exists given that participants had an opportunity to bid for the trips that they preferred—then any resultant differences between the two groups may have affected the results. Third, there was no marker of circadian phase (i.e., body clock time), such as dim-light melatonin onset or minimum of the core body temperature rhythm. Therefore, it was not possible to examine whether differences between the two groups of pilots in terms of their subjective fatigue levels and their capacity to sustain attention were influenced by differences in the degree to which the timing of their body clocks were altered during the short and long layovers.
Reis C, Mestre C, Canhão H (2013)	Cross-sectional study	An amount of 456 commercial airline pilots. mean age was 39.31 (SD 8.39 yr), 442 (96.9%) were men	To identify and quantify the prevalence value of self-reported total and mental fatigue in the community of Portuguese pilots; and to assess and compare the level of fatigue in the two most common types of flight: medium/short-haul and long-haul.	The prevalence values for total and mental fatigue achieved in the Portuguese airline pilots were: 89.3% (FSS > or =4) and 94.1% (FSS > or =4) when splitting the sample in two subsamples, long- and medium/short-haul pilots. Levels of total and mental fatigue were higher for medium/short-haul pilots. The analysis of fatigue levels in each type of aviator showed that medium/short-haul pilots presented the highest levels of total and mental fatigue.	The evaluation instrument (FSS) is a self- report questionnaire, and the study did not attempt to control the timing for each individual response (before, during, or after a flight). This presents inherent subjectivity and limitations. Nonetheless, it is a widely used survey and has been proven effective in the measurement of subjective fatigue. When trying to understand the perception of fatigue in Portuguese pilots, the majority of pilots who responded showed good awareness of his/her level of fatigue. However, when it came to documenting their fatigue to the airline company (using a human factors confidential report), they preferred not to report.
Gander P, Van Den Berg M, Mulrine H, Signal L, Mangie J (2013)	Cross-sectional study	An amount of 14 captains (median age = 56 yr) and 16 first officers (median age = 48 yr), all men	This study tracked circadian adaptation among airline pilots before, during, and after trips where they flew from Seattle (SEA) or Los Angeles (LAX) to Asia (7–9 time zones westward), spent 7–12 d in Asia, and then flew back to the USA.	The findings suggest that two opposing influences were affecting sleep and PVT performance across days in Asia: a progressive circadian adaptation to local time, which meant that the optimal physiological time for sleep increasingly overlapped with local night; and an increase in the amount of duty time during local night, which displaced sleep from the optimal physiological time. Cumulative sleep restriction, particularly in the last few days in Asia, may have contributed to the large rebound in TST seen on the first day at home post-trip. Possible factors accelerating readaptation after the return flight compared with adaptation to Asia time include high homeostatic sleep pressure, no duty constraints on sleep timing, more regular exposure to the day/night cycle, and perhaps stronger social and family cues.	This study has a number of limitations linked to operational factors that were beyond the control of the authors. The unexpected large variability in the trip patterns flown by the participants limited the statistical power of the study as well as the types of analyses that were possible. The uneven distribution of night work across days in Asia was a confounding influence on sleep timing, minimized as far as possible by developing the “physiological” sleep propensity distributions.
Feijó D, Câmara VM, Luiz RR (2014)	Cross-sectional study	778 male commercial airline pilots, 78.33% aged > 35 years	To investigate the association of psychosocial aspect of the work of commercial airline pilots and the prevalence of common mental disorders (CMD).	Multiple logistic regression showed a strong association with highly demanding work and prevalence of CMD, compared to pilots with less demanding work as the reference group (adjusted OR = 29.0). In the final adjusted model, only variables related to workload and physical activity maintained statistically significant associations. The expected CMD prevalence in pilots with highly demanding work, heavy workload, and no regular physical exercise was 39.7%, compared to the subgroup with less demanding work, regular physical exercise, and light work- load, which showed an expected prevalence of 0.4%. Working conditions can be considered potential contributing factors to CMD, with probable impact on flight safety.	Main study limitations were that CMD prevalence was determined by a screening instrument and using the gold standard (psychiatric interview). The investigation has limitations inherent to cross-sectional studies, identifying associations without establishing causal relationships and displaying a selection bias (healthy worker effect) since workers on sick leave were not included in the population assessed. Additionally, non-respondents of the questionnaire (refusal to participate) may have influenced results, either by increasing or decreasing the true prevalence of CMD, or by affecting the association analyses.
Gander PH, Mulrine HM, van den Berg MJ, Smith AA, Signal TL, Wu LJ, Belenky G (2014)	Cross-sectional study	An amount of 237 pilots (97 captains, mean age 55 yr, range 41–63 yr; 132 first officers, mean age 46 yr, range 29–63 yr; 8 in-flight relief pilots, mean age 29 yr, range 27–42 yr)	To investigate the independent contributions to fatigue of flight duration, flight direction, and departure time.	Preflight subjective fatigue and sleepiness were lowest for flights departing 14:00–17:59. Total in-flight sleep was longest on flights departing 18:00–01:59. At TOD, fatigue and sleepiness were higher and PVT response speeds were slower on flights arriving 06:00–09:59 than on flights arriving later. PVT response speed at TOD was also faster on longer flights.	None declared by the authors.
Lawson BK, Scott O, Egbulefu FJ, Ramos R, Jenne JW, Anderson ER (2014)	Cross-sectional study	An amount of 382 participants to the study group (consisting of a multinational cohort of high- performance/high 1G-load aircraft pilots, cargo/passenger aircraft pilots, and trainer aircraft pilots). 57 for the control group (nonaviator military officers of similar age and time in service). 94% male, 18–50 yrs	To elucidate the overall risk and demographic/occupational predictors of neck pain among professional aviators.	Multivariate analysis reveals that the pilot profession is independently predictive of increased occupational neck pain symptoms (OR 1.94, 95% CI 3.72, 1.01). High performance airframes, cargo/passenger airframes, and increasing age (41–50 yrs) were also independent predictors of increased neck pain scores (OR = 3.91, 95% CI 7.10, 2.15; OR = 3.22, 95% CI 5.83, 1.77; OR = 4.00, 95% CI 7.43, 2.15, respectively). In summary, pilot profession, most notably high performance and long-haul cargo/passenger airframes, display an increased risk of neck pain symptoms.	This survey-based study did not include any functional analyses to detail the degree of impairment secondary to the studied occupational exposures. These data points are useful when quantifying specific occupational limitations related to neck pain in the aviator cohort. Future research efforts will include these analyses to determine the effects of targeted muscle strengthening exercises, strain-reducing neck and head mounted equipment, and increased neck support during 1G maneuvers on aviation-related occupational neck pain.
Runeson-Broberg R, Lindgren T, Norbäck D (2014)	Cross-sectional study	An amount of 351 pilots commercial airline (61% captains and 39% first officers), 91.5% male	To assess the prevalence of musculoskeletal symptoms among Swedish commercial pilots, to study the associations between musculoskeletal symptoms, flight length and psychosocial work conditions based on the demands control support model and musculoskeletal symptoms. Another aim was to study differences between job titles among pilots, with respect to musculoskeletal symptoms, psychosocial work environment and investigated associations.	Commercial pilots with high work demands suffered from musculoskeletal symptoms. Pilots on long-haul flights had less elbow symptoms (OR 0.34, 95% CI 0.14–0.85), and women had more hand symptoms (OR 2.90, 95% CI 1.11–7.52). There were associations between high work demands and symptoms from the neck (OR 2.04, 95% CI 1.45–2.88), shoulders (OR 1.46, 95% 1.05–2.03), elbows (OR 1.79, 95% CI 1.10–2.90), and low back (OR 1.42, 95% CI 1.02–1.96) in pilots. Low social support was associated with symptoms from the neck (OR 1.87, 95% 1.35–2.58), shoulders (OR 1.56, 95% CI 1.14–2.14), and low back (OR 1.63, 95% CI 1.18–2.24). Low supervisor support was associated with neck (OR 1.67, 95% CI 1.22–2.27), shoulders (OR 1.38, 95% CI 1.02–1.87), and low back symptoms (OR 1.48, 95% CI 1.09–2.01). Flight captains and first officers showed significant differences in reporting psychosocial factors, and there appeared to be an interaction between job title and psychosocial stress, with respect to musculoskeletal symptoms.	One drawback was the relatively low response rate, even if there were no differences in age and gender between responders and non-responders. There was a higher response rate from captains than from first officers, which could have introduced selection bias, but the magnitude of the difference was relatively small. Another drawback was the cross-sectional study design, which does not allow cause effect associations to be made.
Vejvoda M, Elmenhorst EM, Pennig S, Plath G, Maass H, Tritschler K, Basner M, Aeschbach D (2014)	Cross-sectional study	An amount of 40 commercial short-haul pilots (15 captains and 25 first officers) were studied during a total of 188 flight duty periods, of which 87 started early and 22 finished late. Aged between 21 and 52 years (mean 32 yrs, SD 6 yrs; all males)	To analyzed commercial pilots’ fatigue levels immediately after landing in late-finishing flights and compared them with finishing flights of pilots starting early.	Pilots on late-finishing flight duty periods were more fatigued at the end of their duty than pilots on early starting flight duty periods, despite the fact that preceding sleep duration was longer by 1.1 h. Linear mixed-model regression identified time awake as a preeminent factor predicting fatigue. Workload had a minor effect. Pilots on late-finishing flight duty periods were awake longer by an average of 5.5 h (6.6 versus 1.1 h) before commencing their duty than pilots who started early in the morning. Late-finishing flights were associated with long times awake at a time when the circadian system stops promoting alertness, and an increased, previously underestimated fatigue risk. It is demonstrated that short-haul pilots in the present study experienced moderate to severe fatigue when finishing FDPs late at night. These fatigue levels exceeded those observed after FDPs with early starts, despite the fact that FDP duration was shorter and prior sleep period time was longer.	None declared by the authors.
Zhao R, Xiao D, Fan X, Ge Z, Wang L, Yan T, Wang J, Wei Q, Zhao Y (2014)	Cohort prospective	A group of 305 were chosen randomly by random sampling from all aviators who received a physical examination in 2006 in an aviation medicine examination center in Xi’an Civil Aviation Hospital. All were male civil aviators, mean age was 35.52 ± 7.87 in 2006, 20–55 years	To analyze blood lipid levels, temporal trend, and age distribution of dyslipidemia in civil aviators in China.	The present data indicate that, from 2006 to 2011, the prevalence of high TG and of high LDL-C among civil aviators in China was similar to that in the general population in China as a whole. However, the overall blood lipid levels and dyslipidemia prevalence rates in this cohort were higher than those in the general male population in China. These results indicate that dyslipidemia among civil aviators in China is a serious issue and that, consequently, more attention should be paid to the control and prevention of dyslipidemia in this population.	It should be noted that the number of civil aviators recruited in this study was less than that of some large-scale surveys on blood lipid levels because the target population was quite small relative to the general population and a relatively short study time period was examined. Additionally, there may be some other causes of dyslipidemia in aviators such as hypothyroidism, renal disease, hepatic disease, and diabetes. Nevertheless, there was sufficient power in this study to demonstrate significant trends in blood lipid levels, prevalence rates, temporal trends, and age of dyslipidemia onset among civil aviators in China. Further study with more civil aviators and a longer follow-up period will provide more insight into the epidemiology of dyslipidemia among civil aviators in China.
Palmeira MLS, Marqueze EC (2016)	Cross-sectional study	1198 Brazilian commercial airline pilots, all male and a mean age of 39.2 years (SD = 9.8 years), range 21–67 years	To identify the prevalence and associated factors of overweight and obesity in Brazilian commercial airline pilots.	It was concluded that the prevalence of overweight and obesity among the commercial airline pilots was high and represents a public health problem in this population. Excess weight was associated with time working night-shifts, difficulty relaxing after work, inadequate sleep on days off, having other chronic diseases, and physical inactivity. In this context, nutritional status can be regarded as the result of dynamic and complex interactions promoted by occupational, sleep, and health factors.	Considering the cross-sectional design of the present study, the data found here provide no evidence of causality, only association. However, this research is relevant to describe the characteristics of this population, until then little studied, as well as with such sample representativity. The study had good internal validity given the high sample power (over 80%), rendering it relevant to this professional category as the first study of its kind in the related literature, having included pilots from different airline carriers.
Wu AC, Donnelly-McLay D, Weisskopf MG, McNeely E, Betancourt TS, Allen JG (2016)	Cross-sectional study	1.826 airline pilots. 26.8% age < 41 yrs (76.7% male), 27% age 41–50 yrs (85.6% male), 24.6% 51–60 yrs (87.5% male) and 21.6% age > 60 yrs (97.7% yrs)	To provide a more accurate description of mental health among commercial airline pilots underscoring symptoms related to clinical depression (hereafter also referred to as “depression”).	An amount of 233 (12.6%) of the 1848 airline pilots answering the PHQ-9 met criteria for likely depression. Of the 1430 pilots who reported working as an airline pilot in the last seven days at time of survey, 193 (13.5%) met these criteria. Seventy-five participants (4.1%) reported having thoughts of better being off dead or self-harm within the past two weeks. There was a significant trend in proportions of depression at higher levels of use of sleep-aid medication (trend test z = 6.74, *p* < 0.001) and among those experiencing sexual harassment (z = 3.18, *p* = 0.001) or verbal harassment (z = 6.13, *p* < 0.001).	The inability to draw causal inferences due to the study design was acknowledged. However, the numbers raise concern regarding mental health among pilots. Limitations of this study include potential underestimation of frequencies of adverse mental health outcomes due to less participation among participants with more severe depression compared to those with less severe or without depression. This would lead to downward bias of the true estimate of depression prevalence over the survey period. Conversely, upward bias could occur if participants with underlying mental illness are more likely to participate and complete a survey than those without illness due to participant familiarity with the purpose of the study. Upward bias was probably minimized since participants are less likely to know the focus of the study because the survey covers many topics other than depression or suicidal thoughts. In addition, the survey was not described to participants as a mental health study but as a pilot health study.
Marqueze EC, Nicola ACB, Diniz DHMD, Fischer FM (2017)	Cross-sectional study	1234 pilots working in national or international flights, 97.1% were males, 39.1 years (SD = 9.8 years)	To identify factors associated with unintentional sleep at work of airline pilots.	The prevalence of unintentional sleep while flying the airplane was 57.8%. The factors associated with unintentional sleep at work were: flying for more than 65 h a month, frequent technical delays, greater need for recovery after work, work ability below optimal, insufficient sleep, and excessive sleepiness.	The sample of the study may be under-represented, because only pilots associated with ABRAPAC participated in it. No information was available on the sociodemographic characteristics of those who did not participate in the study, which may also have led to a bias in the results. The cross-sectional design of this study did not allow causal inferences to be made between the variables analyzed.
O’Hagan AD, Issartel J, Nevill A, Warrington G (2017)	Cross-sectional study	An amount of 701 European-registered commercial airline pilots, male and female, aged <25–65 years, captain and first/second officer	To investigate the differences in self-reported depression or anxiety among European-registered commercial airline pilots, and then to further investigate the extent to which these differences could be explained, initially by individual demographic characteristics (e.g., age, position, employment), and subsequently by their experiences of fatigue in the cockpit, experiences of microsleeps in the cockpit, and sleep disturbance due to work schedule.	Differences in self-reported depression or anxiety associated with duty hours were found among European-registered commercial airline pilots. These differences cannot be fully explained by demographic characteristics. Differences in perceived depression or anxiety appear to be explained further by job-related fatigue and sleep disturbance. Due to the detrimental and dangerous influence mental health issues can have on work, flying performance, and thus flight safety, further investigation is essential to determine how to identify, monitor, treat, and reduce factors, which may negatively influence mental health. Although this study assessed whether job-related fatigue and sleep disturbances could explain the differences in perceived depression or anxiety associated with duty hours, the study did not explore which of these factors (e.g., scheduling, circadian rhythms, workload) contribute to the relationship among duty hours, depression, and anxiety.	Several study limitations were identified. First, observations were based on self-report rather than objectively measured sleep and fatigue experiences and ratings of depression or anxiety. Self-report research can be biased by potential misunderstanding of posed questions, social desirability as well as cognitive difficulties associated with recall. Moreover, baseline or clinical levels of depression and anxiety were not pre-determined. Further study would permit researchers to establish a causal relationship between perceived depression or anxiety, duty hours, and experiences of job-related fatigue and sleep disturbance.
van Drongelen A, Boot CR, Hlobil H, Smid T, van der Beek AJ (2017)	Cross-sectional study	An amount of 502 pilots who participated in the MORE energy study. 15.5% aged 21–30 yrs, 28.9% 31–40 yrs, 41.4% 41–50 yrs and 14.1% 51–60 yrs. 93.2% male, 44.6% captain, 38.4% first officer and 16.9% second officer	To determine risk factors for fatigue among airline pilots, taking into account person-, work-, health-, sleep-, and lifestyle-related characteristics.	An amount of 29.5% were classified as being fatigued. Higher age, being an evening type, disturbance of the work-life balance, more need for recovery, a lower perceived health, less physical activity, and moderate alcohol consumption were shown to be risk factors for fatigue.	The results of the study, however, might have been hampered due to selection effects, in case the pilots decided to participate in the MORE energy study because they were more fatigued than their non-participating colleagues. It should also be acknowledged that the study is limited by its cross-sectional nature so that causal relations cannot be established. Another limitation of the study is that it mainly relied on self-reports through online questionnaires, which may have caused misclassification. Finally, considering the homogeneous study population and the specific working situation of airline pilots, it is difficult to generalize the results of the study to other working populations.
Bhat KG, Verma N, Pant P, Singh Marwaha MP (2019)	Cross-sectional study	A total of 1185 pilots were studied. The mean age of the pilots was 34.8 6 13.7 yr (range 18 to 65). There were 1071 men and 114 women	To understand the prevalence of hypertension and obesity in civil aviation pilots and their correlation.	Prevalence of hypertension was 4.1%. Maximum hypertensives were in the 26–35 yr age group. Under the new ACC/AHA guidelines, prevalence of HT was 18.7%. Prevalence of overweight and obesity as per WHO criteria were 39% and 7.3% and, as per Asia Pacific guidelines, were 23.3% and 46.3%, respectively. As BMI increased above 23, risk of developing hypertension or white coat hypertension as per JNC VIII criteria increased by 6.86 times (OR 6.86, 95% CI 0.9–52.58).	None declared by the authors.
Demerouti E, Veldhuis W, Coombes C, Hunter R (2019)	Cross-sectional study	An amount of 1147 pilots were members of the European pilots’ professional association, 91.4% male, mean age was 46.8 years	To uncover the work characteristics (job demands and resources) and the outcomes (job crafting, happiness, and simulator training performance) that are related to burnout among airline pilots.	An amount of 40% of the pilots who participated in the study experienced very high burnout compared to the norms of working populations, whereas 20% could be classified as having high burnout compared to populations under treatment for burnout. In line with the predictions of the job demands–resources theory, burnout was related to the perceptions of job demands and job resources, which highlights the importance of psychosocial work characteristics for the experience of burnout among pilots. Burnout is not only important in its own right as it indicates diminished occupational health. Pilots’ burnout was related to two important outcomes: happiness and simulator training performance. The higher the level of burnout, the less happy the pilots were with their life, which highlights the tremendous implications of burnout for the life of pilots. Burnout was not only related to diminished happiness or wellbeing, but also to an observable, more objective outcome, which is relevant for pilots: performance at the simulator training. Results, however, failed to confirm the direct negative effect of burnout on simulation training performance. Rather, the results suggest that burnout diminishes performance at simulator checks/training because pilots lack the (energy) resources to adjust (craft) their work characteristics such that they fit their preferences and enhance their optimal functioning. These findings have important theoretical and practical implications.	The application of a cross-sectional design to examine presumed causal relationships between the variables represents the first limitation of this study. In a related vein the data are self-reports and therefore could be subject to common method biases, which threaten the robustness of the findings and precludes causal inferences. Consequently, the true associations between the constructs might be weaker than the relationships observed in the study. The findings regarding the performance at simulation checks/training should be interpreted with caution. Due to the nature of the pilots’ work, several of the items used had to be modified or dropped. This resulted in some scales consisting of only two items or in scales with reliabilities lower than the conventional criterion. Although data were successfully collected from a large number of pilots, the sample differs significantly from the population in several respects (gender, age, function level). This means that generalization of the findings to the total population should be done with caution as older captains may experience their work differently than younger first officers.
Albermann M, Lehmann M, Eiche C, Schmidt J, Prottengeier J (2020)	Cross-sectional study	An amount of 693 pilots, 92.2% men, 39.9 yr (SD 8.6)	To determine the point prevalence of acute, subacute and chronic nonspecific low back pain (LBP) in commercial airline pilots on the basis of national guidelines. To identify any additional risk factors and acquire up-to-date cross-sectional data on airline pilots with regard to the point prevalence of nonspecific LBP.	The following point prevalence were found: 8.2% acute, 2.4% subacute, 82.7% chronic LBP; 74.1% of all individuals were suffering from current LBP when answered the questionnaire. A total time spent flying greater than 600 h within the last 12 months was significantly related to acute nonspecific LBP. Individuals with any type of LBP were significantly impaired compared to those unaffected. It was found that German airline pilots suffer more often from current LBP than the general population and have a higher point prevalence of total LBP than their European counterparts.	None declared by the authors.
Arsintescu L, Chachad R, Gregory KB, Mulligan JB, Flynn-Evans EE (2020)	Cross-sectional study	An amount of 90 pilots (eight females) from a short-haul commercial airline, The participants ranged in age between 21 and 54 with an average of 33 yrs (±8) yrs	To determine the relationship between pilot workload, performance, subjective fatigue, sleep duration, number of sectors, and flight duration during short-haul operations.	The findings suggest that a higher self-reported work- load is associated with slower reaction time and higher ratings of fatigue. There were very weak, but significant, correlations between mean pilot workload as measured by NASA-TLX and PVT performance, Samn-Perelli ratings, sleep duration, number of sectors, and flight duration. In addition, each workload scale was significantly correlated with PVT performance and Samn-Perelli fatigue ratings. Pilots experienced more PVT lapses, and their response speed was slower when workload was rated higher. Albeit still weak, the highest correlations of the study were between PVT performance and work- load; workload was rated higher when fatigue was also rated as higher. Pilots reported being frustrated, experiencing time pressure, and making more effort on shorter flights. The stepwise regression showed that Samn-Perelli fatigue, the number of sectors and PVT lapses were significant predictors of workload, although the effect was very small.	Although significant, the correlations in the present study were very weak and the variance explained by the predictors in the regression model was very small. In addition, the study shows relationships between these factors but does not indicate causality. An additional limitation is that the TLX ratings referred to the flight up to the top-of-descent and it didn’t capture landing, which is the most stressful phase of flight. Finally, the flight sectors studied were relatively short due to the short-haul nature of the study, which may explain why flight duration was not a significant predictor in the regression model.

**Table 6 ijerph-20-03401-t006:** Risk of bias of studies assessed by Newcastle-Ottawa tool or Loney criteria, according to study design.

Authors (Year)	Study Design	Quality	Score/Conclusion
Flight Attendants			
Lowden A, Åkerstedt T (1998)	Prospective cohort	Newcastle-Ottawa	4 (Moderate evidence)
Zeeb H, Langner I, Blettner M (2003)	Retrospective cohort	Newcastle-Ottawa	5 (Moderate evidence)
Ballard TJ, Romito P, Lauria L, Vigiliano V, Caldora M, Mazzanti C, Verdecchia A (2006)	Cross-sectional study	Loney criteria	5
Lee H, Wilbur J, Kim MJ, Miller AM (2008)	Cross-sectional study	Loney criteria	5
Agampodi SB, Dharmaratne SD, Agampodi TC (2009)	Cross-sectional study	Loney criteria	3
Castro M, Carvalhais J, Teles J (2015)	Cross-sectional study	Loney criteria	4
Widyanti A, Firdaus M (2019)	Cross-sectional study	Loney criteria	3
Hu CJ, Hong RM, Yeh GL, Hsieh IC (2019)	Cross-sectional study	Loney criteria	7
Pilots			
Samel A, Wegmann HM, Vejvoda M (1997)	Prospective cohort	Newcastle-Ottawa	5 (Moderate evidence)
Gander PH, Gregory KB, Miller DL, Graeber RC, Connell LJ, Rosekind MR (1998)	Cross-sectional study	Loney criteria	3
Goode JH. (2003)	Cross-sectional study	Loney criteria	3
Eriksen CA, Åkerstedt T (2006)	Cross-sectional study	Loney criteria	5
Powell DM, Spencer MB, Holland D, Broadbent E, Petrie KJ (2007)	Prospective cohort	Newcastle-Ottawa	3 (Limited evidence)
Roach GD, Petrilli RM, Dawson D, Lamond N (2012)	Prospective cohort	Newcastle-Ottawa	4 (Moderate evidence)
Reis C, Mestre C, Canhão H (2013)	Cross-sectional study	Loney criteria	5
Gander P, Van Den Berg M, Mulrine H, Signal L, Mangie J (2013)	Cross-sectional study	Loney criteria	5
Feijo D, Luiz RR, Camara VM (2014)	Cross-sectional study	Loney criteria	5
Gander PH, Mulrine HM, van den Berg MJ, Smith AA, Signal TL, Wu LJ, Belenky G (2014)	Cross-sectional study	Loney criteria	4
Lawson BK, Scott O, Egbulefu FJ, Ramos R, Jenne JW, Anderson ER (2014)	Cross-sectional study	Loney criteria	5
Runeson-Broberg R, Lindgren T, Norbäck D (2014)	Cross-sectional study	Loney criteria	5
Vejvoda M, Elmenhorst EM, Pennig S, Plath G, Maass H, Tritschler K, Basner M, Aeschbach D (2014)	Cross-sectional study	Loney criteria	5
Zhao R, Xiao D, Fan X, Ge Z, Wang L, Yan T, Wang J, Wei Q, Zhao Y (2014)	Prospective cohort	Newcastle-Ottawa	5 (Moderate evidence)
Palmeira MLS, Marqueze EC (2016)	Cross-sectional study	Loney criteria	6
Wu AC, Donnelly-McLay D, Weisskopf MG, McNeely E, Betancourt TS, Allen JG (2016)	Cross-sectional study	Loney criteria	6
Marqueze EC, Nicola ACB, Diniz DHMD, Fischer FM (2017)	Cross-sectional study	Loney criteria	6
O’Hagan AD, Issartel J, Nevill A, Warrington G (2017)	Cross-sectional study	Loney criteria	4
van Drongelen A, Boot CR, Hlobil H, Smid T, van der Beek AJ (2017)	Cross-sectional study	Loney criteria	5
Bhat KG, Verma N, Pant P, Singh Marwaha MP (2019)	Cross-sectional study	Loney criteria	6
Demerouti E, Veldhuis W, Coombes C, Hunter R (2019)	Cross-sectional study	Loney criteria	5
Albermann M, Lehmann M, Eiche C, Schmidt J, Prottengeier J (2020)	Cross-sectional study	Loney criteria	5
Arsintescu L, Chachad R, Gregory KB, Mulligan JB, Flynn-Evans EE (2020)	Cross-sectional study	Loney criteria	5
Pilots and Flight attendants			
Ballard TJ, Lagorio S, De Santis M, De Angelis G, Santaquilani M, Caldora M, Verdecchia A (2002)	Prospective cohort	Newcastle-Ottawa	5 (Moderate evidence)
Omholt ML, Tveito TH, Ihlebæk C (2017)	Cross-sectional study	Loney criteria	5
Düz OA, Yilmaz NH, Olmuscelik O (2019)	Cross-sectional study	Loney criteria	3
Goffeng EM, Nordby KC, Tarvainen M, Järvelin-Pasanen S, Wagstaff A, Skare Ø, Lie JÁ (2019)	Prospective cohort	Newcastle-Ottawa	4 (Moderate evidence)
Åkerstedt T, Klemets T, Karlsson D, Häbel H, Widman L, Sallinen M (2021)	Prospective cohort	Newcastle-Ottawa	3 (Limited evidence)

## Data Availability

Not applicable.

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
