# Peer review of "Organizational Risk Factors for Aircrew Health: A Systematic Review of Observational Studies"

_ijerph, 2023, doi:10.3390/ijerph20043401_

Round 1
Reviewer 1 Report
First thank the authors for the work done.
The study entails an interesting discussion and presents a relevant theme however I believe that work has two points of urgent improvement. The abstract and introduction.
The abstract focuses a lot on the quality and quality indicators of the studies used but should focus on the results found in the systematic review of the literature and its implications.
The introduction could also and should be improved, make it more robust, because the conceptual justifications for carrying out the study and its relevance are poor.
Author Response
RESPONSE LETTER
REVIEWER#1
First thank the authors for the work done.
The study entails an interesting discussion and presents a relevant theme however I believe that work has two points of urgent improvement. The abstract and introduction.
We thank the reviewer for this feedback.
The abstract focuses a lot on the quality and quality indicators of the studies used but should focus on the results found in the systematic review of the literature and its implications.
We thank the reviewer for this suggestion, the text has been revised:
Most of the research conducted on risk factors for the work organization of aircrew was carried out in the United States and the European Union and had moderate or low-quality methodology and evidence. However, the findings are homogeneous and allow the most prevalent organizational risk factors for the health of aircrew to be determined, namely high work demand, long hours, and night work. Consequently, the most pervasive health problems were sleep disturbances, mental health disorders, musculoskeletal disorders, and fatigue. Thus, the regulation of the aircrew profession must prioritize measures that minimize these risk factors to promote better health and sleep for these professionals and, consequently, provide excellent safety for workers and passengers.
Then, because of this change, we added at the end of the conclusion these sentences:
Our findings have interesting implications for gaining a better understanding of the main risk factors of aircrew health since this job is significant for society and guides future discussion about the regulation of the aircrew profession. The present review makes it clear that further studies with this professional category need to be carried out, especially longitudinal studies, to understand the causal relationships between occupational risk factors to health.
The introduction could also and should be improved, make it more robust, because the conceptual justifications for carrying out the study and its relevance are poor.
We thank the reviewer. We added this justification on the end of introduction:
However, no study was found in the literature that summarized the findings on risk factors for these professionals. Having a study that synthesizes the main risk factors and its consequences for the aircrew’s health, will be important to be able to think about health polices for these workers, as well as to synthesize what we already know about the subject and what still needs to be researched.
Elaine Cristina Marqueze
On behalf of the authors

Reviewer 2 Report
Review: Organizational risk factors for aircrew health: a systematic review of observational studies
- A brief summary
The article aims to contribute to extant literature in organizational risk factors for aircrew. The primary objective was to analyze the organizational risk factors for aircrew health i.e. (flight attendants and pilots/co-pilots). The secondary objective was to identify the countries in which studies were carried out, with a focus on the quality content of the publications.
I feel the two objectives have been fulfilled to an extent; however, the relevance of the second objective for the study is not clear.
- General concept comments
Study is relevant to the field of study and contemporary.
Article:
Weaknesses of the study:
Please note these are just personal observations and author does not need to include as I’m sure word limits may not allow this.
1. A table for lines 130 to 135 would have been proper to allow for a better appreciation of the works selected and the age range. The narrative is confusing. For example, is the author saying that works conducted in 1998 were only published from 2017-2021? This is not clear.
2. There is no mention or reference to the periods of 2020-2021 when Covid-19 paralyzed air travels for a significant period.
Review:
· The Systematic Literature review method and the way that it is applied in this study is relevant. However, some selected literature used is quite old as the air travel industry has made tremendous progress since 1997. As commented above a table to show the age range of the selected literature would have assisted as the narrative is confusing.
· The gap in knowledge is not clearly identified.
· Relevant references are used.
- Specific comments
The following were observed:
1. “The majority of studies 25 were conducted in the United States and the European Union and had moderate or low methodological and evidence quality”. Lines 25 and 26, repeated in lines 30,31,32.
2. “Therefore, Fundacentro……”, Lines 43-46, is this not recorded as part of the authors? This explanation should be a footnote not in the article.
3. Line 60, “ ……their consequences” on what? Crew health, work performance what exactly?
Author Response
RESPONSE LETTER
REVIEWER#2
- A brief summary
The article aims to contribute to extant literature in organizational risk factors for aircrew. The primary objective was to analyze the organizational risk factors for aircrew health i.e. (flight attendants and pilots/co-pilots). The secondary objective was to identify the countries in which studies were carried out, with a focus on the quality content of the publications.
I feel the two objectives have been fulfilled to an extent; however, the relevance of the second objective for the study is not clear.
We thank the reviewer for this comment.
- General concept comments
Study is relevant to the field of study and contemporary.
We thank the reviewer for this comment.
Article:
Weaknesses of the study:
Please note these are just personal observations and author does not need to include as I’m sure word limits may not allow this.
- A table for lines 130 to 135 would have been proper to allow for a better appreciation of the works selected and the age range. The narrative is confusing. For example, is the author saying that works conducted in 1998 were only published from 2017-2021? This is not clear.
We thank the reviewer for pointing this out because there is an error. The correct sentence is:
The studies involving flight attendants were conducted between 1998- and 2019, 25% of which were published in 2019 (Table 3).
- There is no mention or reference to the periods of 2020-2021 when Covid-19 paralyzed air travels for a significant period.
This is a good point. We did not find any paper discussing this period in our review. Probably, soon new studies will discuss this specific time.
Review:
- The Systematic Literature review method and the way that it is applied in this study is relevant. However, some selected literature used is quite old as the air travel industry has made tremendous progress since 1997. As commented above a table to show the age range of the selected literature would have assisted as the narrative is confusing.
We thank the reviewer for this suggestion. To highlight the years of publications, we have bolded the years in all tables (3, 4, 5 and 6).
- The gap in knowledge is not clearly identified.
We thank the reviewer. We added this gap on the end of introduction:
However, no study was found in the literature that summarized the findings on risk factors for these professionals. Having a study that synthesizes the main risk factors and its consequences for the aircrew’s health, will be important to be able to think about health polices for these workers, as well as to synthesize what we already know about the subject and what still needs to be researched.
- Relevant references are used.
We thank the reviewer.
- Specific comments
The following were observed:
- “The majority of studies 25 were conducted in the United States and the European Union and had moderate or low methodological and evidence quality”. Lines 25 and 26, repeated in lines 30,31,32.
We thank the reviewer. We removed the first sentence (lines 25 and 26).
- “Therefore, Fundacentro……”, Lines 43-46, is this not recorded as part of the authors? This explanation should be a footnote not in the article.
Ok, the sentence has been removed.
- Line 60, “ ……their consequences” on what? Crew health, work performance what exactly?
It is for health. The text has been revised, and we deleted the “commercial”:
Previous studies by Goode [4], Powell et al. [5], Marqueze et al. [6], Goffeng et al. [7], and Pellegrino and Marqueze [8] have highlighted that organizational aspects, such as long working hours, work demands and schedules, numbered among the main risk factors for commercial aircrew health.
Elaine Cristina Marqueze
On behalf of the authors
